# Unsupervised Radar Point Cloud Enhancement Using Diffusion Model as Prior without Paired Training Data

## Abstract

In industrial automation technology, radar is one of the crucial sensors in the machine perception stage. However, due to the long wavelength of radar electromagnetic waves and the limited number of antennas, the angle resolution is limited. Recent advancements have introduced methods that leverage paired LiDAR-radar data for training, achieving notable point enhancement effect. However, the requirement for paired data significantly increases the cost and complexity of model development, limiting model's widespread adoption and scalability. To address this, we propose an unsupervised radar point cloud enhancement algorithm using diffusion model as prior without paired training data. Specifically, our method formulates radar angle estimation recovery into an inverse problem and introduces prior knowledge via a diffusion model when solving it. Experimental results demonstrate that our method achieves high fidelity and low noise performance compared to traditional regularization methods. Compared to paired data training methods, our approach not only delivers comparable performance but also offers greater content control and reduced generation variance. Additionally, it does not require a huge amount of paired data. To the best of our knowledge, our method is the first to enhance radar point cloud by introducing prior knowledge via diffusion model instead of training on paired data.

## 1 Introduction

Radio Detection and Ranging (Radar) technology has been extensively used in robot and related technology. In traffic monitoring, radar systems have been employed for vehicle detection (Palffy et al., 2020) and tracking (Liu et al., 2024; 2023). In the realm of intelligent driving, radar has been integrated with other sensors like LiDAR (Yang et al., 2022) and cameras (Zheng et al., 2023; Xiong et al., 2023) to enhance object detection (Liu et al., 2022; Xiong et al., 2022) and cooperative perception (Huang et al., 2023). However, limited by the number of antennas and hardware noise, the angular resolution of radar is restricted.

To enhance the angular resolution capability of radar, the number of antennas can be increased using MIMO (Multiple Input Multiple Output) techniques (Bliss & Forsythe, 2003). However, how to further improve radar angular resolution under fixed hardware parameters remains a persistent research challenge for scientists. With the advent of deep learning methods, some studies directly use radar points as input (Jin et al., 2023; Kim et al., 2024), employing deep models to fit the mapping function from sparse radar points to dense LiDAR points. However, due to the excessive sparsity of the input information, these approaches exhibit limited performance. To enrich the input information, researchers have used radar signals as input (Prabhakara et al., 2023). When mapped to paired high-resolution LiDAR data, this input signal contains more comprehensive information (Cheng et al., 2021), thereby enhancing the method's performance. Nevertheless, models based on direct mapping from paired data suffer from issues such as insufficient generalization and lack of detail in reconstruction.

To address this, methods based on generative models have been proposed, such as diffusion model (Song et al., 2020; Ho et al., 2020). Methods usually set LiDAR point cloud as target data domain, radar point cloud (Zhang et al., 2024) or range-azimuth heatmap (Wu et al., 2024; Luan et al.,

Figure 1: (a) A diffusion model is trained within the latent space of LiDAR data, where $\mathcal{E}$ and $\mathcal{D}$ are encoder and decoder; (b) Radar point cloud enhancement is achieved by sampling the posterior, which combines the prior knowledge learned from the LiDAR data in (a) with the radar measurements from (c); (c) At each reverse diffusion step, the gradient of the L2 distance between the estimation and measurement is applied to preserve the fidelity of the radar data.

2024) as condition. During the training of the generative model, paired training is also conducted (Rombach et al., 2022), requiring the model to output corresponding LiDAR data when given the radar signal. By learning the distribution and structure of the data, diffusion models based radar point cloud enhancement methods can produce high-quality, diverse new samples, often outperforming direct CNN model mappings in many tasks. Despite this, the aforementioned methods still require one-to-one correspondence between radar and LiDAR data, which limits the model's flexibility and increases the burden of data collection and model training.

To address this issue, this paper proposes a new paradigm for radar point cloud enhancement. By using a pre-trained diffusion model as a prior, the method inputs radar range-azimuth heatmaps and outputs enhanced point clouds with matching LiDAR point cloud density and detail, without the need for additional training. Specifically, our proposed method consists of four steps as illustrated in Fig. 1: Firstly, as shown in (a), the data pre-processing and latent space diffusion model are trained solely on LiDAR data. Secondly, in (b), the reverse process from step (a) is used to start generating random high-quality points. Thirdly, in (c), the radar range-azimuth heatmap is injected at each reverse step of (b) to guide the convergence direction of the generation. Finally, in (d), the method of using the gradient from the radar imaging forward model for control is demonstrated.

The main contributions of this paper include:

- We introduce a novel approach of unsupervised radar point cloud enhancement using diffusion model as prior without paired training data. To the best of our knowledge, this is the first time such a technique has been successfully introduced in the field of radar point cloud enhancement.

- We performed parameter manifold analysis to select optimal hyperparameters when using the proposal method given the radar point cloud enhancement task with a trained latent diffusion model.

- We evaluated the performance of the proposed method on a real-world dataset and compared it with traditional methods. The results show that our method achieves comparable performance in terms of point cloud quality and detail without using paired training data.

## 2 RELATED WORKS

**Discriminative Model-based Methods**: With the advancement of deep learning technology, researchers have been able to leverage the powerful fitting capabilities of neural networks to directly train models that map inputs to outputs. These models are referred to as discriminative models. Some studies have adopted approaches based on radar point input data, where the data is structured in 2D (Jin et al., 2023) or 3D perspectives and then fed into UNet networks (Kim et al., 2024) to train the mapping parameters from radar input to LiDAR output. However, the performance of these approaches is limited by the sparse information contained in the radar points. To improve the results, the input data has been replaced with radar signals, such as time domain signal (Jiang et al., 2023), range-Doppler signal (Cheng et al., 2021; 2022), range-azimuth signal (Prabhakara et al., 2023) or range-Doppler-elevation-azimuth signal (Han et al., 2024; Roldan et al., 2024a;b). This method also employs CNN/Transformer-based network architectures to implement such mappings. Due to the introduction of richer information, this approach achieves better enhancement effects. Nevertheless, these models suffer from significant generalization issues and lack of output details.

**Generative Model-based Methods**: To address the issue of model generalization, methods based on generative models have been proposed. Generative models can achieve the function of randomly generating data within a specific domain by learning the mapping from Gaussian distributions to the statistical distributions of specific data (Song et al., 2020). As the current state-of-the-art generative model, the diffusion model accomplishes this mapping through a process of gradually adding noise to the original data until it becomes Gaussian noise, and then reversing this process via a Markov chain (Ho et al., 2020). During training, the model uses neural networks to learn the parameters of the noise at each step of the noise-adding process. During the denoising process, the neural network is used to progressively remove the noise, generating new data within the target domain (Rombach et al., 2022). Based on this, some studies have set LiDAR as the target data domain for enhanced point clouds and radar as the conditional input (Wu et al., 2024), training various conditioning diffusion models. These studies improve inference performance by adding new consistency constraints (Zhang et al., 2024) during training or by setting biased mapping domains (Luan et al., 2024). However, all of these approaches require well-paired data for training, which severely limits the model's flexibility and wide-range usage. Therefore, this paper proposes an unsupervised radar point cloud enhancement method using a diffusion model as prior without paired training data. Under the inverse problem of radar angular estimation reconstruction that we established, the method utilizes a diffusion model trained in the LiDAR data domain and adds extra gradient constraints (Luo et al., 2020; 2023) from the perspective of inverse problem to achieve radar point cloud enhancement (Rout et al., 2024).

## 3 THE PROPOSED METHODS

In this section, the radar angle measurement model will be described firstly. Then, solving the radar angle estimation inverse problem based on Bayesian theorem will be introduced. Finally, better results by introducing diffusion model is presented.

### 3.1 RADAR ANGLE MEASUREMENT FORWARD MODEL

As depicted in Fig. 2, the electromagnetic waves reflected from object arrives at radar's different antennas in different time. This time shift results in phase shift of the waves (Friedlander et al., 1979; Rosen et al., 2000). Assuming the right antenna as the first antenna. The electromagnetic wave received by $k$-th antenna is described as

$$\vec{\mathbf{E}}(t) = E_0 e^{j\vec{\omega}(t-k\frac{\triangle l}{c})},$$ (1)

where $E_0$ and $\vec{\omega}$ represents the amplification and angular frequency of electromagnetic wave at time $t$. $c$ is the speed of light. $\triangle l$ is the wave traveling length, which can be calculated by

$$\triangle l = d\cos(\theta).$$ (2)

$d$ is the distance between each antenna, which is a constant. Here $\theta$ for different antenna is different. However, We often make the following approximation: the incident electromagnetic wave is parallel to the antenna because the antenna spacing distance is much smaller than the distance from object

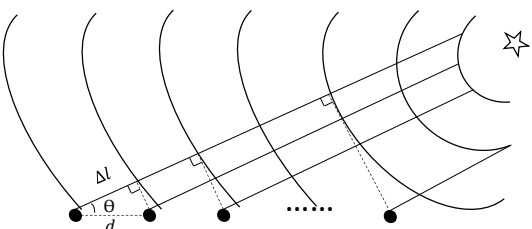

Figure 2: Schematic diagram of radar angle measurement principle. The black dots array represent radar antennas array. The five-pointed stars represent objects detected in space. The distance between object and radar is usually much further than the distance between antennas. Therefore, the incident electromagnetic field is usually considered as a plane electromagnetic wave. It arrives at each antenna at the same angle.

to the radar (Logvin et al., 2002; Cheney & Borden, 2009). Therefore, the signals phase received by antenna array can be expressed as

$$\boldsymbol{S} = \{0, \frac{d\cos\theta}{c}, 2\frac{d\cos\theta}{c}, \cdots, N\frac{d\cos\theta}{c}\}^T. \tag{3}$$

Here $N$ is the number of the antenna. Considering there are $M$ objects separate in space with angles to radar antennas array of

$$\boldsymbol{x} = \{\theta_1, \theta_2, \cdots, \theta_M\}^T. \tag{4}$$

The signal received by antennas array is

$$\boldsymbol{y} = \{\boldsymbol{S}_1, \boldsymbol{S}_2, \cdots, \boldsymbol{S}_M\} = \begin{bmatrix} 0 & 0 & \cdots & 0 \\ \frac{d\cos(\cdot)}{c} & \frac{d\cos(\cdot)}{c} & \cdots & \frac{d\cos(\cdot)}{c} \\ 2\frac{d\cos(\cdot)}{c} & 2\frac{d\cos(\cdot)}{c} & \cdots & 2\frac{d\cos(\cdot)}{c} \\ \vdots & \vdots & \ddots & \vdots \\ N\frac{d\cos(\cdot)}{c} & N\frac{d\cos(\cdot)}{c} & \cdots & N\frac{d\cos(\cdot)}{c} \end{bmatrix} \begin{bmatrix} \theta_1 \\ \theta_2 \\ \vdots \\ \theta_M \end{bmatrix}. \tag{5}$$

Therefore, the radar angle measurement forward model can be simply described as

$$\boldsymbol{y} = A\boldsymbol{x} + \eta, \tag{6}$$

with a noise factor $\xi$. Here, $\boldsymbol{y}, A$ and $\boldsymbol{x}$ are known as measurement, system matrix and unknown state vector. Usually, the environment object such as road or building walls are continuous, each points on such structure is able to reflect electromagnetic wave. Therefore $M$ is infinite theoretically. However, the antenna number $N$ is much less than $M$, thus made Eq. 6 an under-determined Equations. Estimating the unknown state $\boldsymbol{x}$ from the measurement $\boldsymbol{y}$ constitutes an ill-pose inverse problem Eq. 6.

### 3.2 ANGLE ESTIMATION RECONSTRUCTION VIA BAYESIAN ESTIMATION

Normally, when $\boldsymbol{S}$ is acquired by radar antenna array, the angle of specific object can be solved by Fourier analysis. Apply Fourier transformation on $\boldsymbol{S}$, the maximum frequency which is proportional to angle can be solved as

$$f_{max} = \frac{c}{d\cos\theta}. \tag{7}$$

Therefore,

$$\theta = \arccos\frac{df_{max}}{c}. \tag{8}$$

However, limited by the number of antennas, the FFT results are usually in low resolution. When signals of multiple objects which are close each other are acquired at the same time on antenna, the $f_{max}$ of each object would be overlapped. In order to find a more accurate solution, the Bayesian estimation can be applied to the problem described by Eq. 6.

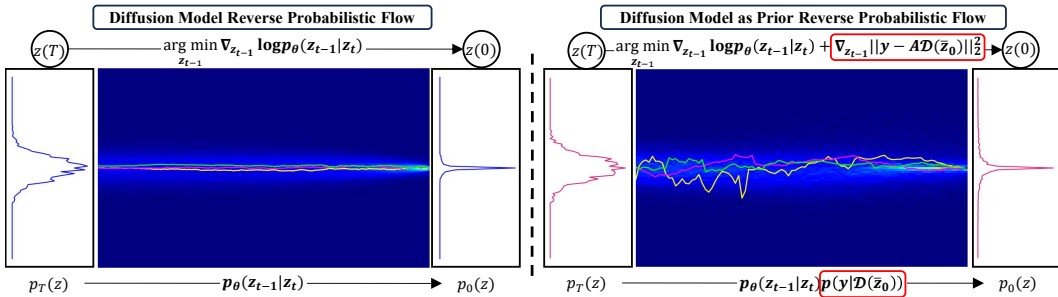

Figure 3: Comparison of reverse probabilistic flow between merely diffusion model (left) and our model using diffusion model as prior (right).

Assuming noise $\eta \sim \mathcal{N}(0, I)$ in Eq. 6, then the likelihood function is depicted as

$$p(\boldsymbol{y}|\boldsymbol{x}) = \mathcal{N}(A\boldsymbol{x}, \boldsymbol{I}). \tag{9}$$

In our case, the measurement $\boldsymbol{y}$ from radar is known, we need to estimate the $\boldsymbol{x}$ with maximum probability from the limited measurement and the likelihood function, that is the posterior probability $p(\boldsymbol{x}|\boldsymbol{y})$. According to Bayesian theorem, the posterior probability $p(\boldsymbol{x}|\boldsymbol{y})$ is proportional to the multiplication of prior probability $p(\boldsymbol{x})$ and likelihood function $p(\boldsymbol{y}|\boldsymbol{x})$ as below:

$$p(\boldsymbol{x}|\boldsymbol{y}) = \frac{p(\boldsymbol{y}|\boldsymbol{x})p(\boldsymbol{x})}{p(\boldsymbol{y})} \propto p(\boldsymbol{y}|\boldsymbol{x})p(\boldsymbol{x}). \tag{10}$$

Because the likelihood function $p(\boldsymbol{y}|\boldsymbol{x})$ as defined in Eq. 9 is knwon, we can apply the Maximum Likelihood Estimation (MLE) method on it directly to find the optimal distribution of $\boldsymbol{x}$. However, this approach can lead to issues such as high variance or sub-optimal solutions. To address these issues, regularization methods that enhance numerical stability, such as L1/L2 regularization (Shkvarko et al., 2016), can be introduced. However, L1 or L2 regularization assumes that the prior probability $p(\boldsymbol{x})$ follows a Gaussian or Laplacian distribution, which offers limited assistance to the solution. Therefore, a diffusion model-based prior is introduced as $p(\boldsymbol{x})$.

### 3.3 Apply Diffusion Model as Prior

In this work, we inference the posterior in Eq. 10 using a diffusion model as prior $p(\boldsymbol{x})$. Traditional gradient descend algorithm applies the calculated gradient when l1 or l2 regularization is used, while diffusion model utilize neural network to predict gradient in each reverse step.

#### 3.3.1 Forward Diffusion Process

In the design of forward diffusion process, the original data $\boldsymbol{x}_0 \sim q(\boldsymbol{x_0})$ will be transit into a Gaussian noise distribution $\boldsymbol{x}_T \sim \mathcal{N}(0, \boldsymbol{I})$. In each step, the diffusion process is described as

$$q(\boldsymbol{x}_t|\boldsymbol{x}_{t-1}) = \mathcal{N}(\boldsymbol{x}_t; \sqrt{1-\beta_t}\boldsymbol{x}_{t-1}, \sqrt{\beta_t}\boldsymbol{I}), \tag{11}$$

where the noise scale factor $\beta_t \in \{0 = \beta_0 < \beta_1 < \cdots < \beta_T = 1\}$. With the diffusion process approaches forward, the proportion of noise increases. In addition, because the forward diffusion process is designed with convolution of Gaussian distribution, the result of step $t$ is able to acquire directly from step 0 follows

$$q(\boldsymbol{x}_t|\boldsymbol{x}_0) = \mathcal{N}(\boldsymbol{x}_t; \sqrt{\bar{\alpha}_t}\boldsymbol{x}_0, (1-\bar{\alpha}_t)\boldsymbol{I}), \tag{12}$$

where $\bar{\alpha}_t = \prod_{i=1}^{t}(1-\beta_i)$. In diffusion model training process, a neural model is used for predicting noise $\eta$ in each diffusion step follows

$$\boldsymbol{x}_t = \sqrt{\bar{\alpha}_t}\boldsymbol{x}_0 + \sqrt{1-\bar{\alpha}_t}\eta, \tag{13}$$

which is a equivalent expression of eq. 12. Therefore, the loss function is constructed as

$$\mathcal{L}_\theta = \|\epsilon - \epsilon_\theta(\boldsymbol{x}_t, t)\|^2. \tag{14}$$

Here the neural network $\epsilon_\theta(\boldsymbol{x}_t, t)$ is designed as the function of noised image $\boldsymbol{x}_t$ and step $t$ with learnable parameters $\theta$.

### 3.3.2 REVERSE DIFFUSION PROCESS

Similar to the forward process, the reverse diffusion process can be described as:

$$p_\theta(\boldsymbol{x}_{t-1}|\boldsymbol{x}_t) = \mathcal{N}(\boldsymbol{x}_{t-1}; \boldsymbol{\mu}_\theta(\boldsymbol{x}_t, t), \boldsymbol{\Sigma}_\theta(\boldsymbol{x}_t, t)). \tag{15}$$

Here the $\boldsymbol{\mu}_\theta(\boldsymbol{x}_t, t)$ and $\boldsymbol{\Sigma}_\theta(\boldsymbol{x}_t, t)$ are mean and variation of step $t$ that parameterized by neural network. Specifically, mean $\boldsymbol{\mu}_\theta(\boldsymbol{x}_t, t)$ is usually provided by neural model as

$$\boldsymbol{\mu}_\theta(\boldsymbol{x}_t, t) = \frac{1}{\sqrt{\alpha_t}} \left( \boldsymbol{x}_t - \frac{\beta_t}{\sqrt{1 - \bar{\alpha}_t}} S_\theta(\boldsymbol{x}_t, t) \right). \tag{16}$$

Here the $S_\theta(\boldsymbol{x}_t, t)$ is what we called as score function that provided by Unet. While for variation is usually be set as

$$\boldsymbol{\Sigma}_\theta(\boldsymbol{x}_t, t) = \frac{1 - \bar{\alpha}_{t-1}}{1 - \bar{\alpha}_t} \beta_t \boldsymbol{I}, \tag{17}$$

which is a non-learned part.

### 3.3.3 ANGLE ESTIMATION BY SAMPLING THE POSTERIOR

In order to solve the problem that illustrated in Eq. 10 jointly with diffusion model, the posterior distribution is represented as

$$p(\boldsymbol{x}|\boldsymbol{y}) \propto p(\boldsymbol{y}|\boldsymbol{x})p_\theta(\boldsymbol{x}). \tag{18}$$

However, to reduce the memory cost of GPU, latent diffusion model is applied here. Therefore, the problem should be modified as

$$p(\boldsymbol{z}|\boldsymbol{y}) \propto p(\boldsymbol{y}|\mathcal{D}(\boldsymbol{z}))p_\theta(\boldsymbol{z}), \tag{19}$$

where $\boldsymbol{z}$ is the latent representation of $\boldsymbol{x}$ which needs to be decoded by $\mathcal{D}(\cdot)$. In each step of iteration, the updating process is described as

$$p(\boldsymbol{z}_{t-1}|\boldsymbol{z}_t, \boldsymbol{y}) \propto p(\boldsymbol{y}|\mathcal{D}(\bar{\boldsymbol{z}}_0))p_\theta(\boldsymbol{z}_{t-1}|\boldsymbol{z}_t) \tag{20}$$

Because $\boldsymbol{x}_0 = \mathcal{D}(\boldsymbol{z}_0)$, but $\boldsymbol{x}_t \neq \mathcal{D}(\boldsymbol{z}_t)$. Therefore, the measurement model should always compute the correct gradient from step 0 rather than from step $t$.

In order to estimate $\boldsymbol{z}_{t-1}$ that maximize the probability in Eq. 20, the maximum a posterior estimation (MAP) method is applied. At first, we can write the logarithmic form of the posterior probability

$$\log p(\boldsymbol{z}_{t-1}|\boldsymbol{z}_t, \boldsymbol{y}) \propto \log p(\boldsymbol{y}|\mathcal{D}(\bar{\boldsymbol{z}}_0)) + \log p_\theta(\boldsymbol{z}_{t-1}|\boldsymbol{z}_t). \tag{21}$$

In order to get the maximum output of Eq. 21, We need to differentiate it and set it to zero

$$\nabla_{\boldsymbol{z}_{t-1}} \log p(\boldsymbol{z}_{t-1}|\boldsymbol{z}_t, \boldsymbol{y}) = 0, \tag{22}$$

then

$$\nabla_{\boldsymbol{z}_{t-1}} \log p(\boldsymbol{y}|\mathcal{D}(\bar{\boldsymbol{z}}_0)) + \nabla_{\boldsymbol{z}_{t-1}} \log p_\theta(\boldsymbol{z}_{t-1}|\boldsymbol{z}_t) = 0. \tag{23}$$

First, the gradient calculation of diffusion model is illustrated as

$$\nabla_{\boldsymbol{z}_{t-1}} \log p_\theta(\boldsymbol{z}_{t-1}|\boldsymbol{z}_t) \propto \tag{24}$$

$$\nabla_{\boldsymbol{z}_{t-1}}(-\frac{1}{2}(\boldsymbol{z}_{t-1} - \boldsymbol{\mu}_\theta(\boldsymbol{z}_t, t))^T \boldsymbol{\Sigma}_\theta(\boldsymbol{z}_t, t)^{-1}(\boldsymbol{z}_{t-1} - \boldsymbol{\mu}_\theta(\boldsymbol{z}_t, t)))$$

$$= -\boldsymbol{\Sigma}_\theta(\boldsymbol{z}_t, t)^{-1}(\boldsymbol{z}_{t-1} - \boldsymbol{\mu}_\theta(\boldsymbol{z}_t, t)).$$

From Eq. 16 we known that $\boldsymbol{\mu}_\theta(\boldsymbol{z}_t, t)$ can be determined by the neural network $S_\theta$ that was embedded in diffusion model. Therefore, the gradient from diffusion model can be determined.

Second, for for calculating gradient from measurement model, the process is illustrated below:

$$\nabla_{\boldsymbol{z}_{t-1}} \log p(\boldsymbol{y}|\mathcal{D}(\bar{\boldsymbol{z}}_0)) = \nabla_{\boldsymbol{z}_{t-1}} \|\boldsymbol{y} - \boldsymbol{A}\mathcal{D}(\bar{\boldsymbol{z}}_0)\|_2^2. \tag{25}$$

However, in step $t$, there is only $\boldsymbol{z}_t$ existing. Therefore, the Tweedie's formula (Song et al., 2023) should be applied to estimate $\bar{\boldsymbol{z}}_0$ as

$$\bar{\boldsymbol{z}}_0 = \frac{1}{\sqrt{a_t}}(\boldsymbol{z}_t - \sqrt{1 - \alpha_t} S_\theta(\boldsymbol{z}_t, t)). \tag{26}$$

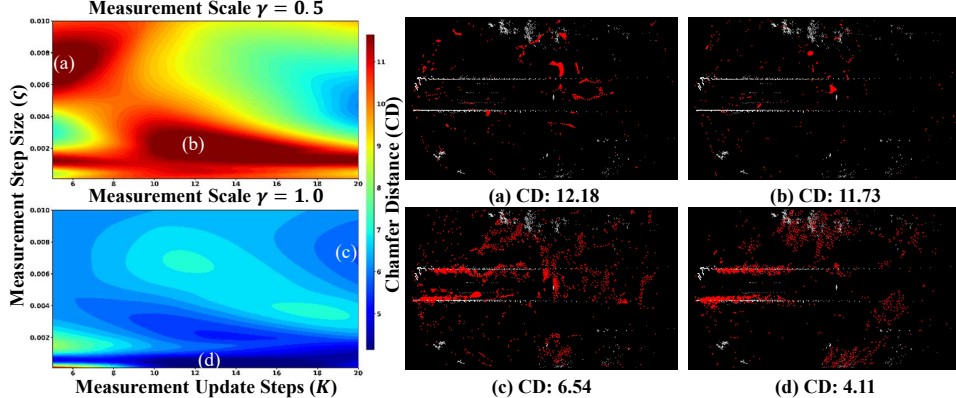

Figure 4: Hyperparameter space manifold analysis of proposed method. The two sub-figures represent model generation results CD metric from measurement scale 0.5 and 1.0. Both of them x-axis represent measurement step number $K$, y-axis represents step size $\zeta$. The output radar points (red) and ground truth LiDAR points (white) are overlaid from (a) to (d) to show their similarity.

At this point, we have obtained the gradients of the two terms of posterior from diffusion model and measurement model. After calculating gradients, the $z_{t-1}$ will be updated. Keep updating $z_t$ with gradient from diffusion model and measurement function, final results can be guided to obtain both high quality of LiDAR-like points and fidelity of the input radar content. Specific process details can refer to Algorithm 1.

A simple comparison of probability flows is also presented in Fig. 3. The left sub-figure illustrates the changes in probability flow during the reverse process when only the diffusion model is used. This sub-figure demonstrates that the trained diffusion model has the capability to recover data from a standard Normal distribution $p_T(z)$ back to the LiDAR distribution $p_0(z)$. In the right sub-figure, the original diffusion model is utilized, incorporating the joint conditions for solving the radar angle estimation inverse problem. The data follows different regression paths but still converges to the target data domain $p_0(z)$. These two output data share the similar distribution, and the right-side process is controlled by the radar data content. Consequently, this results in radar point cloud enhancement outputs high quality results within the LiDAR data domain.

## 4 EXPERIMENTS AND RESULTS

### 4.1 DATASET AND EVALUATION METRICS

**Dataset**: In order to express the effectiveness of our proposed method, an autonomous driving sensing dataset, RADIal dataset (Rebut et al., 2022), is selected. This dataset is a collection of 2-hour of raw data from synchronized automotive-grade sensors (camera, laser, High Definition radar) in various environments (city-street, highway, countryside road) and comes with GPS and vehicle's CAN traces. From the dataset, corresponding LiDAR and radar frames are extracted. The LiDAR data serves as the high-resolution data domain that the radar data is intended to approximate. In order to align with radar range-azimuth data format, LiDAR data are projected into bird-eye's view (BEV) under polar coordinate. Meanwhile, the LiDAR points of ground are filtered because radar is not able to detect ground. Then, a latent diffusion model is trained on this LiDAR dataset as Fig. 1, (a) depicted. On the other branch, radar data is processed as usual. Range FFT, Doppler FFT and Azimuth FFT are applied sequentially. The pre-process output is a radar range-azimuth heatmap in polar coordinate under BEV.

**Evaluation Metrics**: Based on previous research works, Chamfer Distance (CD) is primarily used in our experiments to evaluate the mutual minimum distance between generated 3D points and ground truth 3D points. This metric assesses the similarity of two point sets, with smaller CD values indicating higher similarity. In addition to Chamfer Distance, we also consider other metrics such as Unidirectional Chamfer Distance (UCD), Modified Hausdorff Distance (MHD), Unidirectional Modified Hausdorff Distance (UMHD), and Fréchet Inception Distance (FID).

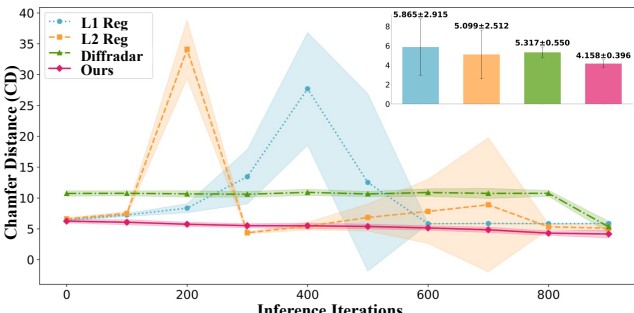

Figure 5: Inference process of radar super-resolution reconstruction using different models. The x-axis shows the inference iterations ranging from 0 to 1000, while the y-axis indicates the Chamfer Distance (CD) between the generated points and the ground truth LiDAR points. The dotted lines represent the mean CD values for each method, and the colored area on each line depict the variation in CD. Up right corner sub-figure depicts the CD mean and variance value at the end iteration.

UCD measures the sum of the squared distances from each point in one set to its nearest neighbor in the other set, but only in one direction. MHD averages the minimum distances rather than taking the maximum, making it less sensitive to outliers. UMHD is a unidirectional version of MHD, calculating the average minimum distance from each point in one set to its nearest neighbor in the other set. FID metric measures the similarity between two datasets of images by comparing the mean and covariance of feature vectors extracted from a pre-trained Inception network. Lower FID scores indicate that the generated images are more similar to the real images, reflecting higher quality and diversity in the generated data.

### 4.2 IMPLEMENTATION DETAILS

First, a Variational Autoencoder (VAE) is trained on an Nvidia-L40 GPU to serve as the initial stage model for the latent diffusion model. Specifically, we employ the VQ-VAE (Van Den Oord et al., 2017) as described in LDM (Rombach et al., 2022). In our experiments, the embedded feature dimension is set to 4, and the quantized channel number is set to 2048. Additionally, the input LiDAR polar BEV data is duplicated three times and concatenated to better leverage the model's capabilities. Once the VQ-VAE has converged, the encoder is used to compress the input LiDAR data into a significantly smaller latent feature map. Subsequently, an unconditioning latent diffusion model is trained on the latent space.

For the inference phase, as illustrated in Fig. 1, the process begins with a randomly initialized Gaussian noise latent feature map. 1000 steps are then taken to recover the final result. As depicted in Algorithm 1, these steps involve alternating between gradient descent based on the diffusion model denoising prior probability and the measurement function likelihood probability. The measurement function is constructed to reflect the actual configuration of the radar sensor used in the RADIal dataset, which employs 84 virtual antennas.

### 4.3 RESULTS

#### 4.3.1 HYPER PARAMETERS MANIFOLD SEARCHING

In our method, there are three hyper-parameters need to be discussed according to the Algorithm 1: Step size ($\zeta$) of measurement model, Step number ($K$) of measurement model, and Measurement scale ($\gamma$). Empirically, above parameters are set as follows: $\zeta \in [0, 0.01]$, $K \in [5, 20]$ and $\gamma \in [0.5, 1.0]$. The hyper parameter space manifold is depicted in Fig. 4.

From this figure, we can observe multiple local optima in the parameter manifold space. Since a smaller CD value indicates that the generated radar point cloud is closer to the ground truth, the overall best performance is achieved at the parameter combination ($d$). For the manifold space slice with a measurement scale ($\gamma$) of 0.5, more measurement steps are needed to compensate for the insufficient regression contribution of each measurement step. For the manifold space slice with a

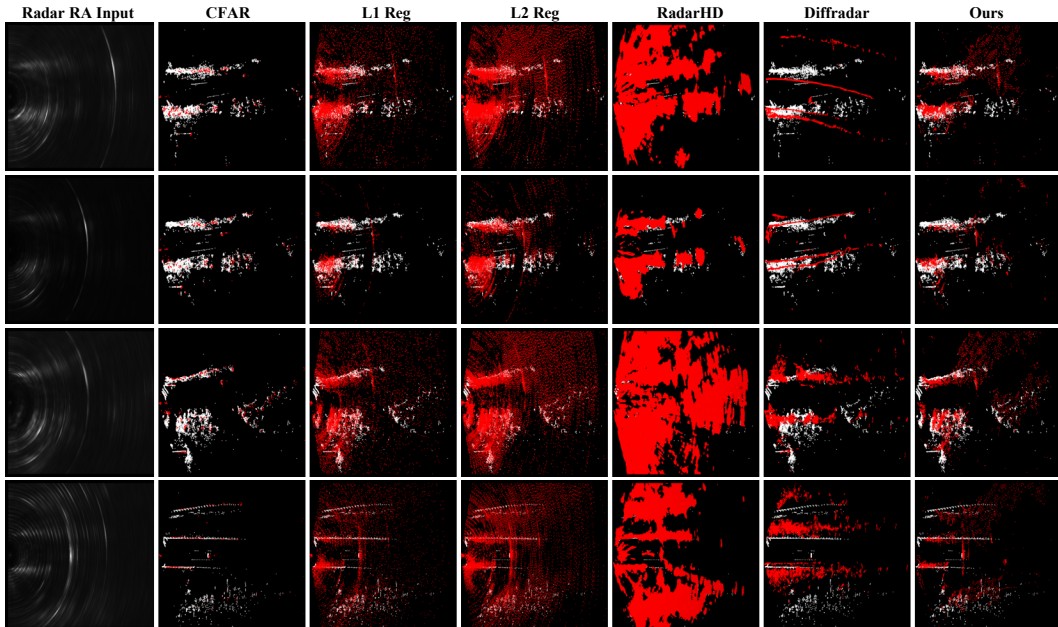

Figure 6: The visualization comparison of the Radial dataset across different methods is presented. The four rows display four randomly selected frames. From the second to the final columns, the super-resolution results from various methods are shown. In each sub-figure, the white points represent the LiDAR ground truth, while the red points indicate the generated super-resolution points.

measurement scale of 1.0, larger values of $K$ and $\zeta$ do not yield better results. Instead, the local optimal parameter combinations are found within $K \in [9, 18]$ and $\zeta \in [0, 0.001]$. This is because excessively large or small values of $K$ or $\zeta$ will guide the model to output results that are either too close to or too far from the radar input. If the output is too close to input, the points enhancement effect is poor, while if it is too far from input, the output becomes uncontrolled. From the visualized parameter combinations (a) to (d) in Fig. 4, we can see that the outputs of (a) and (b) are overly sparse and unrelated to the radar input. Starting from (c), the output shape begins to converge, and noise gradually decreases. Finally, it converges to the global optima at (d), where the CD value is reduced to minima 4.11.

### 4.3.2 INFERENCE VARIATION ANALYSIS

In the framework of iterative optimization methods for solving inverse problems, the solution space typically contains multiple local minima, and the solutions are significantly influenced by random initial values. Therefore, analyzing the variance of the solutions can effectively reflect the stability of the method. As illustrated in Fig. 5, four optimization-based methods are discussed: L1/L2 Regularization (Shkvarko et al., 2016), DiffRadar (Wu et al., 2024), and our proposed method. Five times inference of a same frame radar input are taken to calculate the mean and variance of these methods. It can be observed that, throughout the inference process, our method reaches a lower CD value after the whole inference process. In addition, our method maintain a low variance during the whole inference process together with lower CD values.

### 4.3.3 PERFORMANCE EVALUATION

Statistical performance analysis on the dataset can demonstrate the effectiveness of the proposed method. Table 1 presents the performance metrics of point cloud enhancement results for different methods. Five methods including supervised and unsupervised training methods are selected to compare the performance with our proposed method. Unsupervised methods include CFAR, L1 Reg and L2 Reg. CFAR is a traditional peak extraction method. Other two are methods that original solving radar angle estimation as inverse problem. RadarHD and DiffRadar are two typical super-

Table 1: Statistical Analysis of Point Enhancement Techniques across RADIal Datasets.

| Methods | Supervised Training | $FID_{BEV}\downarrow$ | CD↓ | UCD↓ | MHD↓ | UMHD↓ |
|---|---|---|---|---|---|---|
| CFAR | ✗ | 247.76 | 5.43 | 2.94 | 122.73 | 37.46 |
| L1 Reg (Shkvarko et al., 2016) | ✗ | 225.78 | 5.29 | 3.06 | 137.94 | 31.54 |
| L2 Reg (Shkvarko et al., 2016) | ✗ | 220.23 | 4.82 | 3.52 | **120.68** | 39.51 |
| RadarHD (Prabhakara et al., 2023) | ✓ | 223.34 | **4.43** | 3.17 | 144.13 | 22.90 |
| DiffRadar (Wu et al., 2024) | ✓ | **197.75** | 6.95 | 2.93 | 148.92 | **20.19** |
| Ours | ✗ | 217.87 | 4.60 | **2.71** | 128.36 | 33.88 |

vised approaches that have recently been proposed. The first is a discriminative model using UNet (Ronneberger et al., 2015), while the second is a generative model which is based on conditional-DDPM (Ho et al., 2020). Notably, there are many works on each category, but only RadarHD is open source. DiffRadar is reproduced by authors of this paper.

It can be observed that our method achieves a comparable level of performance on CD-related metrics to supervised learning methods RadarHD. Compared to DiffRadar, the point clouds generated by our method exhibit higher similarity. Fig. 6 shows the results of various radar super-resolution point cloud generation schemes. Compared to the sparse results of CFAR, other schemes provide denser point clouds. However, the L1/L2 regularization schemes introduce a significant amount of noise points, which greatly reduces the accuracy of the generated point clouds. For the RadarHD, the main issue is the complete loss of generated details, indicating the difficulty of constructing details in supervised learning discriminative models. For DiffRadar, while the model indeed has high quality point cloud generation capabilities, it suffers from deficiencies in conditional control. Therefore, it's generated results show a gap in correlation with the input, reducing the precision of this scheme. In contrast, our method generates point clouds with higher density than the CFAR method, lower noise levels than the L1/L2 regularization, more detailed point clouds than RadarHD, and higher control accuracy than DiffRadar. These characteristics make our method a relatively superior approach for radar point cloud enhancement.

## 5 CONCLUSION AND FUTURE WORK

This paper proposes a radar enhancement reconstruction algorithm using diffusion model, which is capable of utilizing the learned prior knowledge from LiDAR data. Bayesian inference for the radar enhancement is formulated, and the samples are drawn from the posterior distribution using the diffusion model as a prior and linear radar imaging equation as a constraint. Through parameter analysis and comparative experiments, the effectiveness and high performance of our method are demonstrated. However, there are still some issues in our method, such as high cost of inference time and multi-modal feature alignment. These issues will be discussed and addressed in future work.

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

# A   APPENDIX

---

**Algorithm 1** Gradient Descent for Angle Estimation with Diffusion Model

---

**Require:** Measurements $\boldsymbol{y}$, total time steps $T$, diffusion coefficient $\lambda_{\text{diff}}$, measurement step size $\zeta$, measurement scale $\gamma$, measurement update iterations $K$

**Ensure:** Refined angle estimate $\hat{\boldsymbol{x}}$

1: $\boldsymbol{z}_T \sim \mathcal{N}(\boldsymbol{0}, \boldsymbol{I})$          ▷ Initialize with random noise
2: **for** $t = T$ to $1$ **do**
3:     // Calculate diffusion model update
4:     $\boldsymbol{\mu}_\theta \leftarrow S_\theta(\boldsymbol{z}_t, t)$          ▷ Predict mean using neural network
5:     $\hat{\boldsymbol{z}}_{t-1} \leftarrow \boldsymbol{z}_t + \lambda_{\text{diff}} \boldsymbol{\Sigma}_\theta(\boldsymbol{z}_t, t)^{-1}(\boldsymbol{\mu}_\theta - \boldsymbol{z}_t)$
6:     // Iterative measurement model update
7:     $\hat{\boldsymbol{z}}_{t-1}^{(0)} \leftarrow \hat{\boldsymbol{z}}_{t-1}$
8:     **for** $k = 1$ to $K$ **do**
9:        $\bar{\boldsymbol{z}}_0^{(k)} \leftarrow \frac{1}{\sqrt{\alpha_t}}(\hat{\boldsymbol{z}}_{t-1}^{(k-1)} - \sqrt{1 - \alpha_t} S_\theta(\hat{\boldsymbol{z}}_{t-1}^{(k-1)}, t))$          ▷ Tweedie's formula
10:        $\boldsymbol{x}_0^{(k)} \leftarrow \mathcal{D}(\bar{\boldsymbol{z}}_0^{(k)})$          ▷ Decode to image space
11:        $\nabla_{\text{meas}}^{(k)} \leftarrow \nabla_{\boldsymbol{z}} \|\gamma \boldsymbol{y} - \boldsymbol{A} \boldsymbol{x}_0^{(k)}\|_2^2$
12:        $\hat{\boldsymbol{z}}_{t-1}^{(k)} \leftarrow \hat{\boldsymbol{z}}_{t-1}^{(k-1)} + \zeta \nabla_{\text{meas}}^{(k)}$
13:     **end for**
14:     $\boldsymbol{z}_{t-1} \leftarrow \hat{\boldsymbol{z}}_{t-1}^{(K)}$
15: **end for**
16: $\hat{\boldsymbol{x}} \leftarrow \mathcal{D}(\boldsymbol{z}_0)$          ▷ Decode final estimate
17: **return** $\hat{\boldsymbol{x}}$

---

