# OpenReview forum: "Unsupervised Radar Point Cloud Enhancement Using Diffusion Model as Prior without Paired Traning Data"
_ICLR.cc/2025/Conference — ICLR 2025 Conference Withdrawn Submission_

### Official Review · Reviewer_16W4 · 2024-10-27

**Soundness:** 3
**Presentation:** 2
**Contribution:** 3
**Rating:** 5
**Confidence:** 3

**Summary:**

This work concerns the enhancement of radar point clouds to develop higher resolution point clouds from original low-resolution point clouds. In general Radar point clouds suffer from low resolution issues, in comparison to LiDAR point clouds that typically have higher resolutions. One approach to solve this problem is the training of deep-learning frameworks that pair LiDAR and Radar data to allow for the creation of higher resolution Radar data. The authors state that this approach tends to be computationally intensive and suffers from the problem of a lack of paired Radar and LiDAR data. The solution that the authors propose is the use of a diffusion model trained with LiDAR data as a prior and not directly requiring the use of paired Radar/LiDAR data for training.

**Strengths:**

The method presented by the work is clearly novel and inventive. The use of a LiDAR prior in the diffusion model appears to be novel to this approach and the problem space (enhancement of Radar images) is fairly important with many key applications. So I feel that the contribution of the work is strong. The description of the work covers the key mathematical details in a high level and I feel that the reproducibility of the work is also clear. Others knowledgeable in the field should be able to follow the math and reproduce the work.
The experimental justification is also fairly strong, clearly showing the advantages of the work over the prior art and justifying the conclusions.

**Weaknesses:**

The primary weakness of the work is the grammar and spelling and the formatting of the text. The authors introduce numerous spelling and grammatical errors that make it unclear at times what the precise meaning of certain sentences are. Figures are also at times very distant from the associated text making it difficult to follow along with the work without constantly flipping from page to page.

**Questions:**

Q1: Figure 1 is quite difficult to read with the images 1(a) being difficult to read and understand. The caption also seems insufficient. Is the first image input LiDAR data and the second a mapping to Radar coordinates? Please explain more fully.
Q2: In Section 2, the authors state "However, the performance of these approaches is limited by the sparse information contained in the radar points. To improve the results, the input data has been replaced with radar signals, such as time domain signal (Jiang et al., 2023),
range-Doppler signal (Cheng et al., 2021; 2022), range-azimuth signal (Prabhakara et al., 2023) or range-Doppler-elevation-azimuth signal (Han et al., 2024; Roldan et al., 2024a;b).". This is confusing. What is meant by "input data has been replaced with radar signals"? Isn't the input a radar signal? Please clarify.
Q3: In line 150, "As depicted in Fig. 2, the electromagnetic waves reflected from object arrives at radar’s different
antennas in different time." should be "As depicted in Fig. 2, the electromagnetic waves reflected from the object arrive at the radar’s different antennas at different times. "
Q4: In line 197, "Usually, the environment object such as road or building walls are continuous, each points on such structure is able to reflect electromagnetic wave." should be "Usually, environmental objects such as road or building walls are continuous, each points on such structure is able to reflect electromagnetic wave."
Q5: In line 240, "knwon" should be "known".
Q6: In line 250, "when l1 or l2 regularization" should be "when L1 or L2 regularization"
Q7: In line 258, "With the diffusion process approaches forward, the proportion of noise increases." I am not sure what is meant by this sentence. Please consider rephrasing. Perhaps: "When the diffusion process is run forward, the proportion of noise increases."
Q8: Please define the term $\epsilon$ in equation (14).
Q9: Line 280 states "While for variation is usually be set as". I believe that the authors are referring to the covariance matrix in this case. To avoid confusion, I suggest rewording as: "While the covariance matrix is usually set as". Also, what is meant by "usually" in this case? Please clarify under which conditions equation (17) holds and which conditions it does not hold.
Q10: In line 315, "From Eq. 16 we known that" should be "From Eq. 16 we know that".
Q11: In line 318, "Second, for for calculating gradient" should be "Second, for calculating gradient"
Q12: Figures are often excessively far from the text describing them. For example, Figure 5 appears on the top of page 8, but isn't described until the bottom half of page 9.

---

### Official Review · Reviewer_zYPf · 2024-10-27

**Soundness:** 3
**Presentation:** 3
**Contribution:** 2
**Rating:** 5
**Confidence:** 4

**Summary:**

- Summary: This paper proposes an unsupervised method for radar point cloud enhancement using a diffusion model as prior, without requiring paired radar-LiDAR training data. The key idea is to formulate radar angle estimation recovery as an inverse problem and introduce prior knowledge through a diffusion model trained only on LiDAR data.

- Main Contribution: Propose a novel unsupervised framework that leverages diffusion models as prior for radar point cloud enhancement, eliminating the need for paired training data.

**Strengths:**

Good Originality:
- The paper proposes a novel perspective of using diffusion models as prior knowledge for radar point cloud enhancement without requiring paired training data. I think this is novel.

Good Quality:
- The methodology is theoretically well-grounded, combining Bayesian inference framework with diffusion models.
- The experimental evaluation is conducted against both traditional methods (CFAR, L1/L2 regularization) and recent deep learning approaches (RadarHD, DiffRadar)

Good Clarity:
- The overall structure of the paper is logical and well-organized

**Weaknesses:**

**Major Weaknesses:**

1. **Insufficient Validation of Generalization Capability - A Critical Concern**
   - The paper's fundamental contribution lies in its unsupervised approach that eliminates the need for paired training data
   - However, the primary benefit of such an approach - enhanced generalization ability - is not properly validated
   - The paper critically lacks:
     * Cross-dataset evaluation (e.g., training diffusion model on one LiDAR dataset and testing on different radar datasets)
     * Cross-scenario validation within the same dataset
     * Analysis of performance under varying environmental conditions
   - This significant omission raises serious doubts about the practical value of the proposed method

> I like the idea of this paper. If the authors could provide proper demonstration of generalization capability (e.g., showing that a diffusion model trained on a larger LiDAR dataset performs well across different radar datasets), I will significantly improve my score.

2. **Unconvincing Experimental Results**
   - The qualitative results reveal a concerning level of noise in the generated point clouds
   - Visual inspection shows more noise points compared to both CFAR and DiffRadar
   - The quantitative metrics fail to demonstrate clear advantages over existing methods

> I suggest the authors to present the results more fairly, for example using fair colors and brightness for mmWave radar and lidar. The current color scheme somewhat hides this shortcoming.

3. The motivation for latent space reconstruction is inadequately justified, being reduced to mere GPU memory considerations rather than methodological benefits

**Minor Weaknesses:**

1. **Technical Presentation and Methodology Issues**
   - The radar angle measurement model contains several technical inaccuracies:
     * "Wave traveling length" is incorrectly used instead of the standard term "path difference"
     * Equation 7's "maximum frequency" lacks proper theoretical foundation or citation
     * The derivation of the angle estimation formula is not clearly explained
   - The connection between generative models and improved generalization lacks proper theoretical justification:
     * The paper claims generative models enhance generalization without explaining the underlying mechanism
     * The relationship between diffusion models and distribution alignment is not properly addressed

2. **Writing Quality and Clarity Concerns**
   - The methodology section suffers from inconsistent tense usage, switching between "will be", "will be", and "is"
   - Multiple instances of imprecise technical descriptions:
     * The statement about radar angular resolution limitations oversimplifies the physical constraints
     * The term "content control" in the abstract is not properly defined or explained
   - Mathematical notation lacks consistency:
     * L1/L2 should be written as L_1/L_2 following standard notation
     * Several equations contain undefined or poorly defined variables

**Questions:**

Please see Weaknesses

---

### Official Review · Reviewer_YEPP · 2024-11-03

**Soundness:** 3
**Presentation:** 4
**Contribution:** 3
**Rating:** 5
**Confidence:** 3

**Summary:**

This paper presents an unsupervised method for enhancing radar point clouds by leveraging a diffusion model as a prior, eliminating the need for paired training data with LiDAR.

**Strengths:**

1) Quality: Figures are well-designed and look good.
2) Performance: experiments show its comparable or superior performance to both supervised and unsupervised methods.

**Weaknesses:**

1) Efficiency:
- The authors note that the current method has a high inference time cost due to the iterative nature of the diffusion process.
- Although this trade-off is partially offset by the removal of the need for paired data, the model’s practical applicability in real-time applications may be limited due to the inference process.
2) Experiment:
- Although the authors perform a parameter manifold analysis, additional ablations could be beneficial.
- Specifically, exploring the impact of different diffusion model architectures or the influence of specific regularization techniques (e.g., L1 or L2) on performance would provide further insights into the robustness of the proposed approach.

**Questions:**

1) Could the authors provide further details on potential optimizations for the inference process to improve efficiency, especially for real-time applications?
2) How does the performance of the proposed method vary with different diffusion model architectures, and could alternative designs reduce the inference time without compromising quality?
3) Have the authors considered applying this method in dynamic scenarios, where both radar and LiDAR data might contain temporal variations?

---

### Official Review · Reviewer_etx8 · 2024-11-04

**Soundness:** 1
**Presentation:** 1
**Contribution:** 2
**Rating:** 3
**Confidence:** 4

**Summary:**

The paper proposes an unsupervised approach to radar point cloud enhancement using diffusion priors, intended to avoid reliance on paired training data.

**Strengths:**

- The problem setting is interesting, as radar point cloud data is significantly lower in both quality and quantity compared to image data. Applying an enhancement method to such data could help improve its quality.

**Weaknesses:**

- The paper is not well-written, and the claims are unpolished. For example, the authors claim, "Unsupervised Radar Point Cloud Enhancement Using Diffusion Model as Prior without Paired Training Data." However, in the abstract, they state: "Compared to paired data training methods, our approach not only delivers comparable performance but also offers greater content control and reduced generation variance. Additionally, it does not require a huge amount of paired data." I am very confused by these contradictory statements. Furthermore, even after careful reading, I cannot find any information about how the authors train their models, i.e., what the input and output of the model are, and what the objective function is.

- The novelty is limited. Although the problem setting is interesting, the authors simply combine the latent diffusion model with radar angle estimation.

- Concerns about its soundness. I believe diffusion priors can play a role in solving the radar enhancement problem; however, since there is only limited information about how the authors train their method (L. 410–412), I am wondering what kind of diffusion priors the authors are using. They simply state: "Subsequently, an unconditioned latent diffusion model is trained on the latent space." Are the authors training the diffusion model from scratch? If yes, how do they define the diffusion prior? Section 3.3 should explain this, yet it merely introduces the vanilla diffusion equations.

**Questions:**

See weaknesses

---

### Note · Authors · 2024-11-25

I have read and agree with the venue's withdrawal policy on behalf of myself and my co-authors.